# Genomes of cultivated and wild *Capsicum* species provide insights into pepper domestication and population differentiation

Feng Liu[1,9], Jiantao Zhao[1,2,9], Honghe Sun[2,3], Cheng Xiong[1], Xuepeng Sun[2,7], Xin Wang[2,8], Zhongyi Wang[1], Robert Jarret[4], Jin Wang[1], Bingqian Tang[1], Hao Xu[1], Bowen Hu[1], Huan Suo[1], Bozhi Yang[1], Lijun Ou[1], Xuefeng Li[5], Shudong Zhou[5], Sha Yang[5], Zhoubing Liu[1], Fang Yuan[1], Zhenming Pei[1], Yanqing Ma[1], Xiongze Dai[1], Shan Wu[2]✉, Zhangjun Fei[2,6]✉ & Xuexiao Zou[1]✉

Pepper (*Capsicum* spp.) is one of the earliest cultivated crops and includes five domesticated species, *C. annuum* var. *annuum*, *C. chinense*, *C. frutescens*, *C. baccatum* var. *pendulum* and *C. pubescens*. Here, we report a pepper graph pan-genome and a genome variation map of 500 accessions from the five domesticated *Capsicum* species and close wild relatives. We identify highly differentiated genomic regions among the domesticated peppers that underlie their natural variations in flowering time, characteristic flavors, and unique resistances to biotic and abiotic stresses. Domestication sweeps detected in *C. annuum* var. *annuum* and *C. baccatum* var. *pendulum* are mostly different, and the common domestication traits, including fruit size, shape and pungency, are achieved mainly through the selection of distinct genomic regions between these two cultivated species. Introgressions from *C. baccatum* into *C. chinense* and *C. frutescens* are detected, including those providing genetic sources for various biotic and abiotic stress tolerances.

The genus *Capsicum*, a member of the nightshade family Solanaceae and commonly known as pepper or paprika, originated in South and Central America and includes around 35 species[1]. In 2021, the global cultivation of peppers exceeded 3.68 Mha, with a total production of 36.29 and 4.84 million tons of green and dry peppers, respectively (FAOSTAT; http://faostat.fao.org/). *C. annuum* var. *annuum* is the most commonly planted *Capsicum* species worldwide and contains both pungent and non-pungent (sweet) pepper fruits. Pungent pepper fruits are not only used as food and nutraceutical resources but also widely used in traditional medicines as pain relief and cancer

[1]Engineering Research Center for Germplasm Innovation and New Varieties Breeding of Horticultural Crops, Key Laboratory for Vegetable Biology of Hunan Province, College of Horticulture, Hunan Agricultural University, Changsha, China. [2]Boyce Thompson Institute, Ithaca, NY, USA. [3]Plant Biology Section, School of Integrative Plant Science, Cornell University, Ithaca, NY, USA. [4]U.S. Department of Agriculture-Agricultural Research Service, Plant Genetic Resources Conservation Unit, Griffin, GA, USA. [5]Institute of Vegetable Research, Hunan Academy of Agricultural Science, Changsha, China. [6]U.S. Department of Agriculture-Agricultural Research Service, Robert W. Holley Center for Agriculture and Health, Ithaca, NY, USA. [7]Present address: College of Horticulture Science, Zhejiang A&F University, Hangzhou, China. [8]Present address: Department of Vegetable Crops, College of Horticulture and Forestry, Huazhong Agricultural University, Wuhan, China. [9]These authors contributed equally: Feng Liu, Jiantao Zhao. ✉e-mail: sw728@cornell.edu; zf25@cornell.edu; zouxuexiao428@163.com

inhibitors, reflecting the presence of a diverse array of bioactive compounds[2].

Pepper has been domesticated in Central and/or South America since around 7000 BC, thus it represents one of the most ancient domesticated crops[3]. The five cultivated *Capsicum* species are *C. annuum* var. *annuum* domesticated from *C. annuum* var. *glabriusculum*, *C. baccatum* var. *pendulum* domesticated from *C. baccatum* var. *baccatum*, *C. chinense*, *C. frutescens* and *C. pubescens*. They belong to three distinct clades: *C. annuum* var. *annuum*, *C. chinense* and *C. frutescens* in the Annuum clade, *C. baccatum* var. *pendulum* in the Baccatum clade and *C. pubescens* in the Pubescens clade[3]. *C. pubescens* has flowers with purple corollas and dark brown or blackish seeds, while *C. baccatum* var. *pendulum* has flowers with white corollas and yellow or golden spots and beige- or tan-colored seeds, and *C. baccatum* var. *baccatum* has flowers with white to off-white corollas and corolla spots. Flowers of the cultivated *C. frutescens* and *C. chinense* have corollas that are typically off-white but more often are pale green[4], while the corollas of *C. annuum* var. *annuum* are white (rarely violet). *C. chinense* is diversified in terms of fruit shape, size and color, while *C. frutescens* fruits are usually small and elongated. Fruits of *C. chinense* are typically quite pungent, with a strong fruit-like flavor[5] and a distinct aroma. Fruits of *C. frutescens* are also pungent. Within the Annuum clade, there is a wide range of overlapping morphological characteristics[6]. For instance, *C. frutescens* shows some characteristics such as fruit shape, color and fruit position similar to those of *C. chinense*.

The domestication of peppers has been suggested to have occurred in multiple centers of origin[7,8]. However, previous large-scale genetic and genomic studies of pepper mainly focused on one single domesticated species, *C. annuum* var. *annuum*[9–11]. Therefore, knowledge of phylogenetic relationships, genetic diversity, population differentiation, convergent and divergent selections, and introgression among the various domesticated and wild *Capsicum* species remains limited. Recent rapid advances in sequencing technologies and abundant available germplasm resources of *Capsicum* have made it feasible to perform large-scale genome analyses of diverse *Capsicum* species to evaluate their genetic diversity and decipher genomic regions and genes associated with the evolution and domestication of peppers.

In this study, we first assemble high-quality genomes of three pepper accessions from the Annuum, Baccatum and Pubescens clades, using PacBio long reads and high-throughput chromosome conformation capture (Hi-C) maps, and construct a graph pan-genome from these three genome assemblies. We then perform genome resequencing of 500 accessions that cover all five domesticated species and the important wild progenitors and relatives and construct a single-base resolution variation map of *Capsicum* using the graph pan-genome as the reference. Our comprehensive population genomic analyses provide insights into population differentiation, convergent and divergent domestication, and introgression of the *Capsicum* species, and pinpoint candidate genes associated with important agronomic traits such as fruit shape and flavor, flowering time and biotic/abiotic stress responses. This study provides comprehensive and valuable genomic resources for facilitating future functional studies and molecular breeding of the *Capsicum* species.

## Results
### Reference genome assembly and graph pan-genome construction
We selected three lines, *C. annuum* var. *annuum* Zhangshugang, *C. baccatum* var. *pendulum* PI 632928, and *C. pubescens* Grif 1614, from the Annuum, Baccatum and Pubescens clades, respectively, for sequencing and assembly of pepper reference genomes. Using PacBio long reads, Illumina short reads and Hi-C data (Supplementary Fig. 1 and Supplementary Table 1), we generated chromosome-scale genome assemblies for Zhangshugang, PI 632928, and Grif 1614, with the contig

N50 lengths of 35.42 Mb, 27.70 Mb and 46.20 Mb, respectively (Supplementary Fig. 2, Supplementary Data 1 and Supplementary Table 2). Evaluation of base accuracy and completeness of the genome assembly using Merqury[12] and BUSCO[13] indicated the high quality of these three genome assemblies (Supplementary Data 1 and Supplementary Table 3). The three assemblies showed an overall high collinearity, although some large inversions and genome rearrangements were observed between them (Supplementary Fig. 3).

The percentages of repetitive sequences were 78.14% in the Zhangshugang genome, 87.21% in the PI 632928 genome and 89.25% in the Grif 1614 genome (Supplementary Table 4). Combining evidence from ab initio predictions, transcript mapping and protein homology, a total of 33,688, 32,830 and 33,398 protein-coding genes were predicted in the Zhangshugang, PI 632928 and Grif 1614 genomes, respectively. Gene BUSCO analysis indicated that 94.0–97.7% of the core conserved plant genes were completely captured in the predicted genes of these three genomes, compared with only 77.0–86.7% in the predicted genes of previously published pepper genomes (Supplementary Data 1).

To improve read mapping and variant calling and to provide a more comprehensive reference to access the genetic diversity of the *Capsicum* species, we constructed a graph pan-genome from these three high-quality chromosome-scale genome assemblies. Using the Zhangshugang genome as the reference, small insertions/deletions (indels; 1–20 bp) and structural variants (SV; >20 bp) were identified in PI 632928 and Grif 1614 based on the whole-genome alignments. These identified variants in PI 632928 and Grif 1614 were integrated into the Zhangshugang genome, resulting in a graph pan-genome that contained a total of 1635.8 Mb sequences not present in the Zhangshugang genome. Of these sequences, 780.9 Mb and 853.5 Mb were uniquely from PI 632928 and Grif 1614, respectively, and 1.38 Mb were from both accessions (Supplementary Data 2).

### Phylogeny and population structure of *Capsicum*
In this study, we first selected 1296 accessions representing nine *Capsicum* species collected from 94 countries or regions for shallow whole-genome sequencing (-1×) (Supplementary Data 3). Using SNPs called from the shallow whole-genome sequencing data, we constructed a core collection using GenoCore[14], which comprised 500 accessions and captured -97.5% of the total genetic diversity in the 1296 *Capsicum* accessions. Accessions in the core collection were mainly from Central and South America (Fig. 1a and Supplementary Data 4). These 500 accessions were further sequenced at an average depth of 14.7×, resulting in a total of 22.3 terabases of sequence data (Supplementary Data 4). By mapping the sequencing reads to the graph pan-genome, a total of 100,111,632 high-quality SNPs and 5,306,979 small indels were identified in the core collection. The SNPs included 430,414 that caused non-synonymous changes and 27,251 that resulted in start/stop codon gains or losses (Supplementary Table 5).

Species misidentifications and ambiguity among accessions are not uncommon in herbaria[15] and *Capsicum* germplasm collections[16]. It is well known that *C. annuum*, *C. chinense* and *C. frutescens* show evidence of parallel evolution for various plant and fruit characteristics[17]. The overlap in morphological characteristics of these three species has led to the recognition of a *C. annuum* complex[18]. In most instances, these taxa are effectively identified using a botanical key[4] utilizing species-specific characteristics such as those illustrated in Heiser and Smith[19]. While these characteristics have proven useful in most instances, they are subject to some degree of quantitative variation as affected by environmental and/or genetic influences. Hence, molecular evidence has become increasingly useful in clarifying questionable taxonomic classifications[16]. Of the 500 accessions in the core collection, 62 were originally unidentified as to species (*C.* spp.), and 9 were unidentified as to variety (*C. baccatum* var. *baccatum* or *C. baccatum* var. *pendulum*) (Supplementary Data 4). Based on the phylogenetic relationship and population structure inferred from

28,980 linkage disequilibrium (LD)-pruned SNPs at the fourfold degenerate (4DTv) sites (Supplementary Fig. 4), combined with the manual review of data provided in the U.S. National Plant Germplasm System (NPGS) and images of the fruit of these accessions grown in Changsha, Hunan province, China, all 62 previously unclassified accessions were unambiguously assigned to a specific species and/or variety group within a species, and nine *C. baccatum* accessions without variety classification were assigned to an appropriate variety group (Supplementary Data 4). An additional 80 accessions were found to have been incorrectly classified and were subsequently reassigned to the correct species (*n* = 47) or variety group (*n* = 33). It is notable that 10 accessions, all from Mexico, previously classified as wild *C. annuum* var. *glabriusculum* (9) or unclassified (1), were clustered within and

shared a similar genetic structure with the cultivated *C. annuum* var. *annuum* accessions, suggesting that these accessions are possibly feral forms (Fig. 2b and Supplementary Data 4).

After taxonomy correction, the 500 accessions were classified into seven *Capsicum* species, *C. pubescens* (*n* = 38), *C. chacoense* (*n* = 17), *C. baccatum* (*n* = 118), *C. annuum* (*n* = 112), *C. galapagoense* (*n* = 1), *C. frutescens* (*n* = 99) and *C. chinense* (*n* = 115). *C. annuum* was further divided into cultivated (*C. annuum* var. *annuum*; *n* = 90, including 10 possible feral accessions) and wild (*C. annuum* var. *glabriusculum*; *n* = 22) forms. *C. baccatum* was also divided into cultivated (*C. baccatum* var. *pendulum*; *n* = 109) and wild (*C. baccatum* var. *baccatum*; *n* = 9) groups (Fig. 1a, b). The placement of the seven species in the phylogenetic tree was consistent with the previously reported

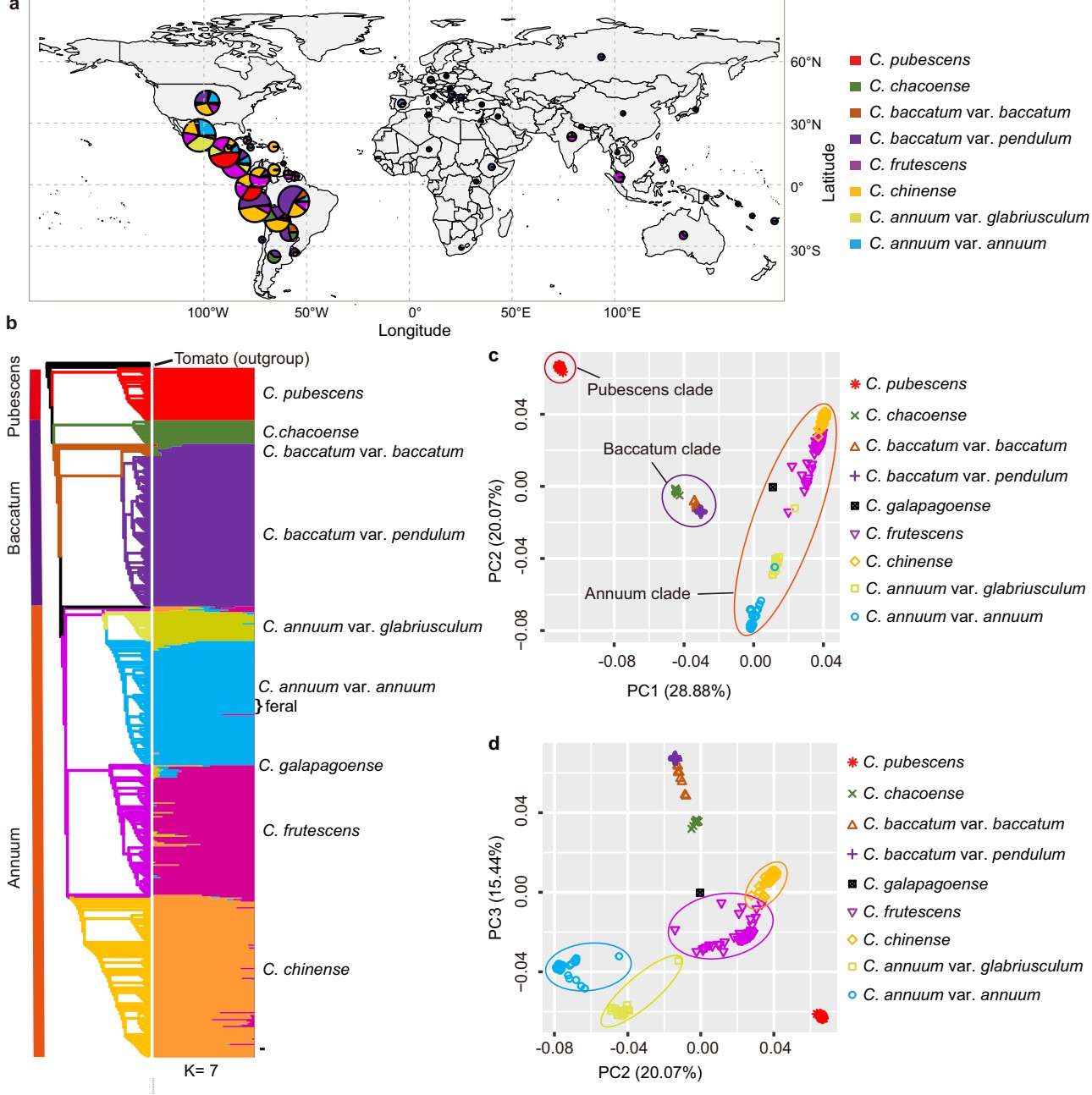

**Fig. 1 | Phylogeny and population structure of *Capsicum* accessions.**
**a** Geographical distribution of the core pepper accessions (after taxonomy correction). The size of the pie corresponds to the total number of accessions from the selected geographical region. For *C. galapagoense*, only one accession with

unknown geographic information was available and is not presented on the map. **b** Maximum likelihood phylogenetic tree and population structure of the core pepper accessions. **c**, **d** Principal component (PC) analysis, PC1 versus PC2 (**c**) and PC2 versus PC3 (**d**), of the core pepper accessions.

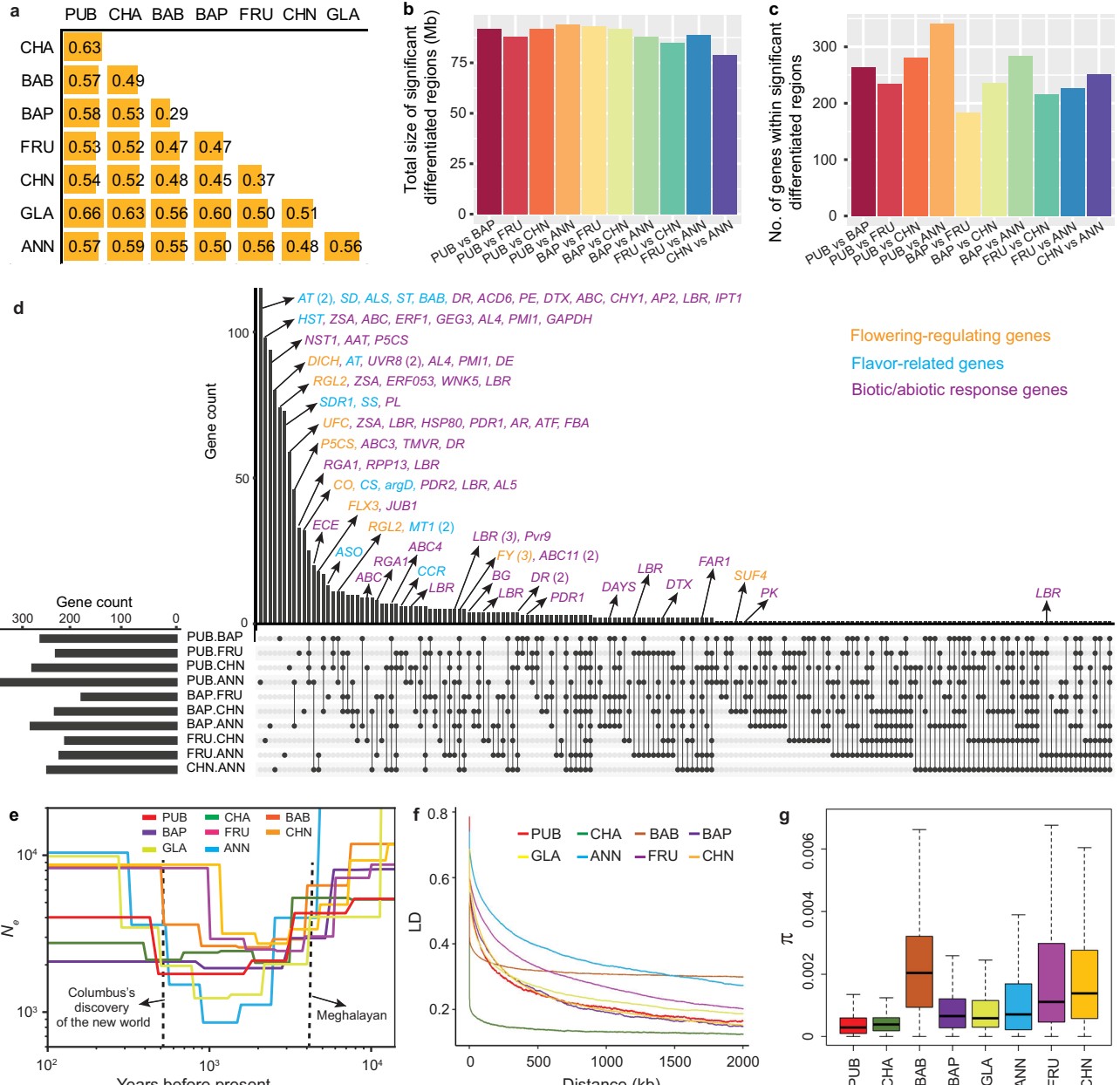

**Fig. 2 | Population differentiation and history of *Capsicum* species. a** Fixation index ($F_{ST}$) between different pepper populations. For each comparison, all $F_{ST}$ values generated from 100 permutations were lower than the empirical $F_{ST}$ value. **b** Total size of highly differentiated genome regions (top 1% $F_{ST}$) among the five domesticated *Capsicum* species. **c** Number of genes within the highly differentiated regions. **d** UpSet plot of common and unique genes within the differentiated regions among the five domesticated *Capsicum* species. **e** History of the effective population size of different pepper groups. **f** Linkage disequilibrium (LD) decay pattern of different pepper groups. **g** Boxplots of nucleotide diversity (π) for different pepper groups. For each box plot, the lower and upper bounds of the box indicate the first and third quartiles, respectively, and the center line indicates the median. The numbers of windows (*n*) for PUB, CHA, BAB, BAP, GLA, ANN, FRU and CHN are 28,893, 28,672, 28,736, 29,061, 29,007, 29,141, 29,242 and 29,189, respectively. Note, π values could be overestimated due to the sub-sampling using GenoCore. PUB, *C. pubescens*; CHA, *C. chacoense*; BAB, *C. baccatum* var. *baccatum*; BAP, *C. baccatum* var. *pendulum*; FRU, *C. frutescens*; CHN, *C. chinense*; GLA, *C. annuum* var. *glabriusculum*; ANN, *C. annuum* var. *annuum*. Detailed gene information is provided in Supplementary Data 5.

phylogeny[1], with *C. pubescens* (Pubescens clade) sister to the other six species, followed by *C. chacoense*, *C. baccatum* var. *baccatum* and *C. baccatum* var. *pendulum*, forming the Baccatum clade, and *C. galapagoense*, *C. frutescens*, *C. chinense*, *C. annuum* var. *glabriusculum*, and *C. annuum* var. *annuum*, forming the Annuum clade (Fig. 1b). Population structure analysis revealed seven distinct lineages, which was highly consistent with the phylogenetic tree (Fig. 1b). Principal component analysis (PCA) revealed three primary clusters representing the Pubescens, Baccatum and Annuum clades (Fig. 1c, d), with PC2 clearly separating the four groups, *C. frutescens*, *C. chinense*,

*C. annuum* var. *annuum*, and *C. annuum* var. *glabriusculum*, in the Annuum clade (Fig. 1d).

A total of 38 possible hybrids were identified in these pepper accessions and all were from the Annuum clade, including five from *C. annuum* var. *annuum*, three from *C. annuum* var. *glabriusculum*, 10 from *C. chinense* and 20 from *C. frutescens* (Supplementary Data 4). This result is consistent with the cross-compatibility observed between species in the Annuum clade[20]. These possible hybrids and the 10 possible *C. annuum* var. *annuum* feral accessions were excluded from the downstream population analyses.

## Population differentiation of *Capsicum* species

We studied genome-wide population differentiation using the fixation index ($F_{ST}$) among the five domesticated species groups, *C. pubescens*, *C. baccatum* var. *pendulum*, *C. frutescens*, *C. chinense* and *C. annuum* var. *annuum*, and the three wild groups, *C. chacoense*, *C. baccatum* var. *baccatum* and *C. annuum* var. *glabriusculum*. *C. baccatum* var. *baccatum* and *C. baccatum* var. *pendulum* had the lowest level of genome-wide population differentiation ($F_{ST} = 0.29$) (Fig. 2a). $F_{ST}$ values were larger than 0.37 for all other comparisons, supporting strong population differentiation among *Capsicum* species and independent domestications of the different cultivated species.

The five domesticated species are distributed in specific and unique environments[3], have distinct aroma profiles[21,22] and exhibit a broad spectrum of biotic and abiotic stress tolerances[23]. To better understand the genetic basis of these distinct characteristics, we identified highly differentiated genomic regions (top 1% $F_{ST}$ windows). The total sizes of highly differentiated genomic regions were similar among different pairwise species comparisons (Fig. 2b), while substantially more genes were found in the differentiated regions between *C. pubescens* and *C. annuum* var. *annuum* (Fig. 2c). Though most of the genes were specific to single comparisons between the five domesticated species, genes involved in biotic/abiotic responses were commonly detected in more than one comparison (Fig. 2d and Supplementary Data 5).

Flowering time is a complex trait that reflects the adaptation of plants to their corresponding growing environments. The timing of flowering differs among the five domesticated *Capsicum* species. *C. pubescens* requires the longest time to flower, followed by *C. baccatum* var. *pendulum*, *C. chinense* and *C. frutescens*, whereas *C. annuum* var. *annuum* requires the shortest amount of time. Genes involved in flowering time were found in highly differentiated genomic regions among the domesticated species, such as those homologous to Arabidopsis Suppressor of FRI 4 (*SUF4*) and FLX Expressor 3 (*FLX3*) and *FY* that encodes an RNA 3′ processing protein involved in the regulation of floral transition (Supplementary Data 5). Notably, genes homologous to *SUF4*, *FLX3* and *FY* were identified in highly differentiated regions in multiple comparisons. These genes provide insights into flowering time variation in the *Capsicum* species.

Capsaicinoids contribute greatly to the distinct spicy flavor of fruits of different *Capsicum* species[22,24]. Genes involved in the biosynthesis of capsaicinoids were identified in the highly differentiated regions in comparisons among the five domesticated species, including one encoding the cinnamoyl-CoA reductase (*CCR*; *Caz01g35570*) between *C. frutescens* and *C. pubescens* and between *C. chinense* and *C. pubescens* (Fig. 2d and Supplementary Data 5). Volatiles are important flavor-related compounds that impact consumer preferences[25]. Compounds such as alcohols and aldehydes are commonly found in fruits of *C. annuum* var. *annuum*, *C. chinense* and *C. frutescens*, but their profiles vary among different species[26]. Genes regulating alcohol and aldehyde biosynthesis, such as those encoding alcohol dehydrogenases, were found in the identified differentiated genomic regions. Non-volatile compounds such as sugars were also highly correlated with flavor in sweet pepper[27] and showed intra- and interspecies variation in *Capsicum*[28]. Genes encoding sucrose synthases and sugar transporters were identified in the differentiated genomic regions (Fig. 2d and Supplementary Data 5). These flavor-related genes showed dramatic differences in expression at different fruit development stages (Supplementary Fig. 5). Further functional studies of these genes could improve our understanding of flavor development and variation among domesticated peppers and conceivably lead to the development of more flavorful fruit.

A number of genes involved in biotic/abiotic stress responses were identified in differentiated regions among the domesticated species (Supplementary Data 5). Notably, two tandemly duplicated disease resistance genes, *Caz09g00020* and *Caz09g00030*, were detected in five pairwise comparisons among the five domesticated species. These genes may underlie the resistances found in different pepper groups and could serve as potential targets in pepper resistance breeding.

## Population history of *Capsicum* species

Dynamics of the effective population size ($N_e$) can provide insights into the impacts of past environmental factors and human domestication events on the demographic history of crops[29]. We found that *Ne* for all *Capsicum* species and variety groups decreased dramatically after the Last Glacial Maximum (LGM) period (33 to 19 ka), which profoundly affected Earth's climate and environment[30]. The *Ne* then rapidly recovered to a plateau starting approximately 4000 years ago when Earth entered the Meghalayan age (4.2 ka to present)[31] (Fig. 2e), possibly due to environmental changes and domestication events of the *Capsicum* species. The *Ne* of *C. annuum* var. *annuum* was the lowest among all the groups until recently, when it became the largest. The increase in the *C. annuum* var. *annuum* population size began around 1000 years ago and increased rapidly after Columbus' discovery of the Americas which led to further migration and the secondary global diversification of pepper[3]. The intense domestication and selection of *C. annuum* var. *annuum* might have led to its slowest genome-wide linkage disequilibrium (LD) decay compared to other groups (Fig. 2f) and a significantly lower level (adjusted $P < 0.05$) of nucleotide diversity ($\pi = 1.19 \times 10^{-3}$) compared to the other two domesticated species in the Annuum clade, *C. frutescens* ($\pi = 2.02 \times 10^{-3}$) and *C. chinense* ($\pi = 1.96 \times 10^{-3}$) (Fig. 2g).

## Convergent and divergent domestication of *Capsicum* species

The principal breeding targets of domesticated *Capsicum* species include fruit yield and quality (such as shape and pungency), plant architecture and resistance to biotic and abiotic stresses[23]. Fruits of the two domesticated peppers, *C. annuum* var. *annuum* and *C. baccatum* var. *pendulum*, were significantly longer and wider and had a significantly higher level of morphological diversity than those of their corresponding wild progenitors (Fig. 3a, b), suggesting possible convergent domestication in *C. annuum* var. *annuum* and *C. baccatum* var. *pendulum* leading to larger fruit sizes and more variable fruit shapes.

We searched for genome-wide domestication signals by comparing the domesticated species, *C. annuum* var. *annuum* and *C. baccatum* var. *pendulum*, with their wild progenitors, *C. annuum* var. *glabriusculum* and *C. baccatum* var. *baccatum*, respectively. A total of 529 and 615 selective sweeps were identified for *C. annuum* var. *annuum* (Fig. 3c and Supplementary Data 6) and *C. baccatum* var. *pendulum* (Fig. 3d and Supplementary Data 7), respectively, with a cumulative size of 176.51 Mb (5.84% of the pepper genome) and 181.95 Mb (6.02%). Notably, only 11.17 Mb of genomic regions (containing 68 genes) were found to be under selection in both *C. annuum* var. *annuum* and *C. baccatum* var. *pendulum* (Fig. 3e), while 165.34 Mb (1553 genes) and 170.78 Mb (890 genes) were specifically selected in *C. annuum* var. *annuum* and *C. baccatum* var. *pendulum*, respectively.

Fruit size/weight is among the most important traits targeted during pepper domestication. Genes and QTLs associated with fruit size and weight detected in selective sweeps of *C. annuum* var. *annuum* were largely different from those detected in *C. baccatum* var. *pendulum* (Fig. 3c, d and Supplementary Data 8). One of the strongest domestication signals detected in *C. annuum* var. *annuum* was on chromosome 3 that contained fruit weight QTL *Han3.1*[32], while the strongest signals in *C. baccatum* var. *pendulum* included one on chromosome 2 that overlapped with *Han2.1*[32] and one on chromosome 9 that overlapped with fruit size/weight QTLs *Fwd/Fwg*[33]. Our results suggest that domestications of *C. annuum* var. *annuum* and *C. baccatum* var. *pendulum* leading to increased fruit size/weight were likely achieved through the selection of unique genomic regions.

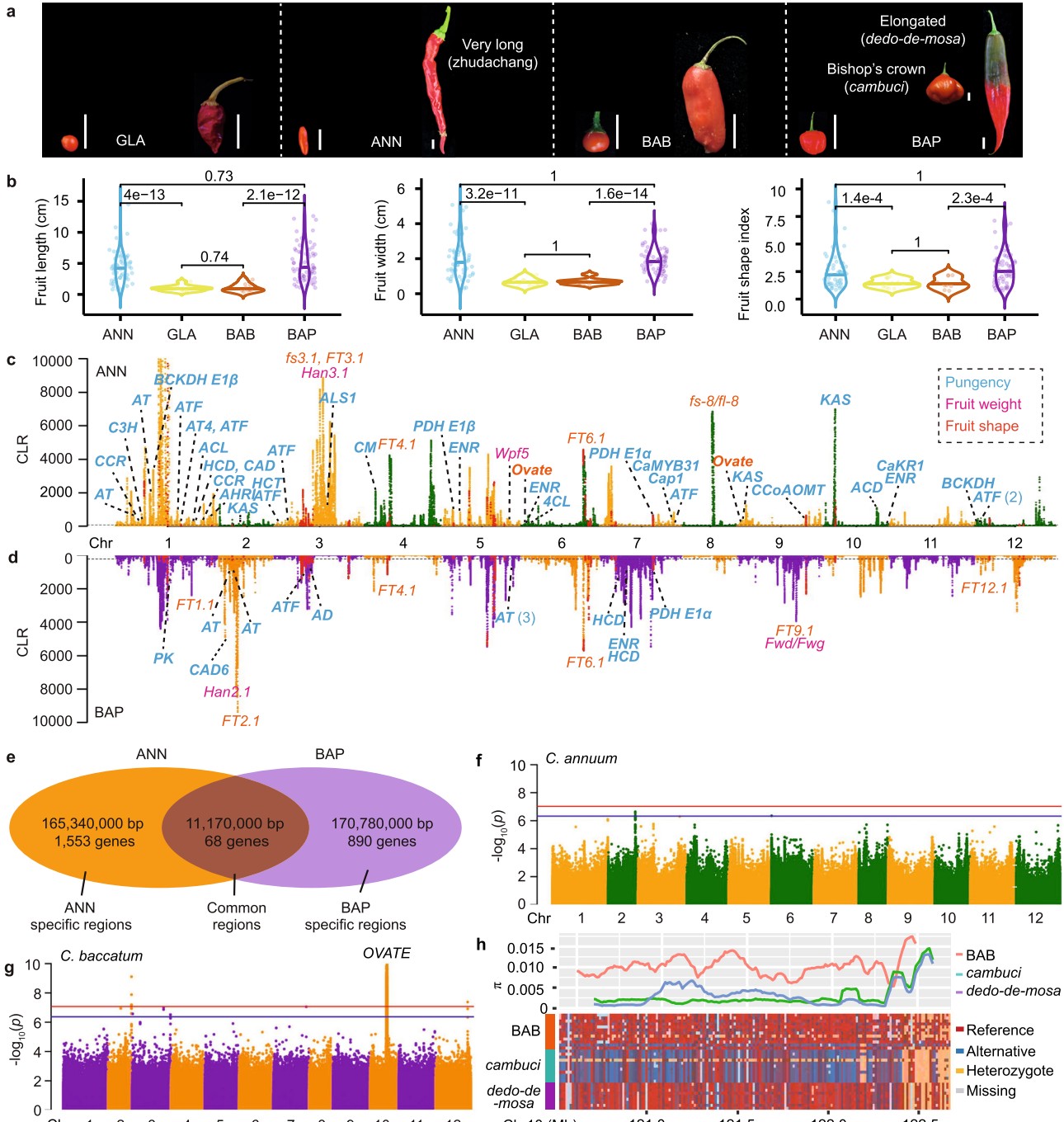

**Fig. 3 | Convergent and divergent domestication of *C. annuum* var. *annuum* and *C. baccatum var. pendulum*. a** Fruits of representative wild and domesticated *C. annuum* and *C. baccatum* accessions with diverse shapes. Vertical bar represents 1 cm. **b** Fruit length, fruit width and fruit shape index in wild and domesticated *C. annuum* and *C. baccatum accessions*. Significant difference between groups was assessed using the two-tailed Student's test. **c, d** Selective sweeps in *C. annuum* var. *annuum* (**c**) and *C. baccatum* var. *pendulum* (**d**). Fruit weight, shape, and pungency-related genes (in bold) and QTLs are presented in pink, orange, and cyan, respectively. Dashed horizontal lines indicate the top 5% Composite likelihood ratio (CLR) scores. Red dots correspond to selective sweeps identified in both *C. annuum* var. *annuum* and *C. baccatum* var. *pendulum*, and orange, dark green, dark orange and purple dots indicate CLR scores on different chromosomes. **e** Common and specific

genome regions and genes under selection during *C. annuum* var. *annuum* and *C. baccatum* var. *pendulum* domestication. **f, g** Manhattan plots of genome-wide association studies of fruit shape of *C. annuum* (**f**) and *C. baccatum* (**g**) accessions. Red and blue horizontal lines indicate the Bonferroni-corrected significance thresholds of GWAS at α = 0.05 and α = 1, respectively. **h** Nucleotide diversity (π) and allele conservation of *C. baccatum* var. *baccatum* and *C. baccatum* var. *pendulum* accessions with fruit shape index >4 (the elongated *dedo-de-moça* type) and those with fruit shape index <1.2 (the *cambuci* type). GLA, *C. annuum* var. *glabriusculum*; ANN, *C. annuum* var. *annuum*; BAB, *C. baccatum* var. *baccatum*; BAP, *C. baccatum* var. *pendulum*. Detailed information on genes and QTLs is provided in Supplementary Data 6. Source data are provided as a Source Data file.

Fruit shape QTLs such as *FT4.1* and *FT6.1* (ref. 34) were found under selection during the domestication of both *C. annuum* var. *annuum* and *C. baccatum* var. *pendulum*, whereas a number of other QTLs were uniquely selected, e.g., *fs3.1* and *FT3.1* only in *C. annuum* var. *annuum* and *FT1.1*, *FT2.1*, *FT9.1* and *FT12.1* only in *C. baccatum* var. *pendulum* (Fig. 3c, d). Two genes (*Caz06g07310* and *Caz09g04240*) encoding OVATE family proteins, which have been reported in tomato to regulate fruit shape[35], were found in the selected regions in *C. annuum* var. *annuum* but not in *C. baccatum* var. *pendulum*. These results indicate that, like fruit weight, the domestication of fruit shape in *C. annuum* var. *annuum* and *C. baccatum* var. *pendulum* was also mainly achieved through the selection of different genomic regions. Our core *Capsicum* collection displayed diverse fruit shapes. The longest and the most elongated pepper accession, Zhudachang, was found in *C. annuum* var. *annuum* (Fig. 3a). *C. baccatum* var. *pendulum* has two popular types, the bell-shaped *cambuci* mainly cultivated and consumed in the southwest of Brazil, and the elongated *dedo-de-moça* in the southern and southeastern regions of Brazil[36] (Fig. 3a). Genome-wide association studies (GWAS) of fruit length and width in *C. annuum* and *C. baccatum* did not detect significant association signals (Supplementary Fig. 6). However, GWAS of fruit shape index in *C. annuum* identified a significant association signal on chromosome 2 between 160,915,869 bp and 161,006,859 bp, harboring five genes (Fig. 3f; Supplementary Table 6). GWAS analysis of fruit shape index in *C. baccatum* identified two highly significant signals, one on chromosome 2 between 172,704,945 bp and 172,706,325 bp, and the other on chromosome 10 between 120,586,110 bp and 122,585,250 bp (Fig. 3g). No genes were annotated in the associated region on chromosome 2, while the region on chromosome 10 contained four genes, including one (*Caz10g08850*) encoding an OVATE family protein (Supplementary Table 6). The nucleotide diversity of the *Caz10g08850* genomic region was much lower in both *C. baccatum* var. *pendulum* groups with accessions having a fruit shape index >4 (the elongated *dedo-de-moça* type) and those having a fruit shape index <1.2 (the *cambuci* type) compared to *C. baccatum* var. *baccatum* (Fig. 3h), suggesting the fixation of different alleles of *Caz10g08850* in different pepper types during pepper breeding to meet consumers' preferences.

Wild peppers are usually pungent, while various pungency levels have been selected for during pepper domestication and modern breeding for different culinary uses. A total of 36 and 14 genes involved in capsaicin biosynthesis were found in selective sweeps of *C. annuum* var. *annuum* and *C. baccatum* var. *pendulum*, respectively (Fig. 3c, d and Supplementary Data 8), with only two (*Caz03g22890* and *Caz07g10470*) of them overlapping. These results indicated that various pungency levels in *C. annuum* var. *annuum* and *C. baccatum* var. *pendulum* were mainly achieved through the selection of different genes in the capsaicin biosynthetic pathway during their independent domestications.

**Introgressions from the Baccatum clade to the Annuum clade**
Introgressing resistance (*R*) genes from closely related wild or cultivated species into modern commercial cultivars is an important and frequently used breeding strategy. For instance, *Mi-1*, conferring resistance to root-knot nematode in tomato, was introgressed from its distantly related wild species *Solanum peruvianum*[37,38]. Among the *Capsicum* species, one significant gene flow event from the Baccatum clade to a subgroup of the Annuum clade, including *C. frutescens* and *C. chinense* was detected using TreeMix[39] (Fig. 4a and Supplementary Fig. 7). The significance of this gene flow event was further supported by the ABBA-BABA test[40] (D = 0.21; *P*-value = 2.55 × 10⁻⁶) (Fig. 4b). The genome-wide average degree of introgression indicated by *fd* was 1.71% and the proportion of genome introgression (PGI) was 1.51%. The strongest introgression signals were detected on chromosomes 6, 7 and 12 (Fig. 4c). The region at the end of chromosome 6 (Chr06:242,250,001-246,150,000; top 1% *fd* as the cutoff,

corresponding to *fd* = 11.38%) contained 72 genes (Supplementary Data 9), including 17 *R* genes, of which 15 formed two clusters with 6 and 9 genes, respectively (Fig. 4d). One gene in the first cluster, *Caz06g28230*, homologous to tomato *Mi-1* (ref. 41), is known to confer nematode resistance in pepper (denoted as *CaRKNR*)[42]. Further scanning of all accessions in the Annuum clade for genomic regions similar to those in the Baccatum clade revealed the presence of a Baccatum introgression in *C. chinense* and *C. frutescens*, which was rarely found in *C. annuum* (Fig. 4e). This introgression led to increased nucleotide diversity levels (Fig. 4d) and stronger LD (Fig. 4f) in this region, in *C. chinense* and *C. frutescens* compared to *C. annuum* var. *annuum*, possibly due to lack of recombination. Genes known to contribute to biotic and abiotic stress tolerance were also identified in the introgressed regions on chromosomes 7 and 12 (Supplementary Fig. 8 and Supplementary Data 9). These results together suggest that the introgression from the Baccatum clade to *C. chinense* and *C. frutescens* could be a consequence of the introduction of resistance traits.

## Discussion
The high-quality chromosome-scale reference genomes and the graph pan-genome of three domesticated pepper species, and the single-base resolution genome variation map of various cultivated and wild pepper species developed in the present study represent a great complement to the recently released pepper reference genomes, pan-genomes and the variation map that mainly focused on one species, *C. annuum*[9–11,43], providing additional valuable resources for pepper breeding, and evolutionary and comparative studies within *Capsicum* and beyond. Large-scale pepper germplasm sequencing reported here serves as an important complementary approach to morphological checking for accurately assigning taxonomy information to plant materials in *Capsicum* germplasm collections, an accomplishment also demonstrated in a previous study[16]. Deep resequencing of 500 *Capsicum* accessions in the core collection captures genetic variation among the five domesticated and three closely related wild species. Phylogenetic and population structure analyses support the independent domestications of the five cultivated *Capsicum* species. Examination of the genetic basis of population differentiation provides insight into key variations of flowering time, fruit aroma and stress tolerance among the five domesticated pepper species (Fig. 5), which helps understand their adaptations to different environments and the impacts of convergent/divergent domestication in shaping the diversity within the *Capsicum* genus in response to consumers' preferences and needs.

Fruit size and shape are among the most dramatic domestication traits of *Capsicum* species. Selective sweeps identified in *C. annuum* var. *annuum* and *C. baccatum* var. *pendulum*, two different pepper domesticated species, indicate that shared selection goals such as for enlarged fruit size and elongated fruit shape were mainly achieved by selection of different genomic regions, despite that a few genomic regions overlapping with known fruit shape QTLs have been selected in both cultivated pepper species (Fig. 5). Combined with GWAS, an OVATE family protein gene, *Caz10g08850*, was identified to be highly associated with fruit shape and has been under selection in different types of *C. baccatum* var. *pendulum* with contrasting fruit shapes. OVATE family proteins have been functionally demonstrated to control fruit shape in plants[35]. Alleles of *Caz10g08850* could be utilized in breeding or gene editing programs to create novel shapes to meet consumers' preferences. Capsaicinoids are the specialized metabolites in *Capsicum* species that determine the pungency. Different genes in the capsaicinoid biosynthesis pathway have been selected in *C. annuum* var. *annuum* and *C. baccatum* var. *pendulum*. In summary, the present study demonstrates that genomic regions selected in different *Capsicum* species for the most important breeding targets, e.g., fruit size, shape and pungency, are largely different, consistent with their

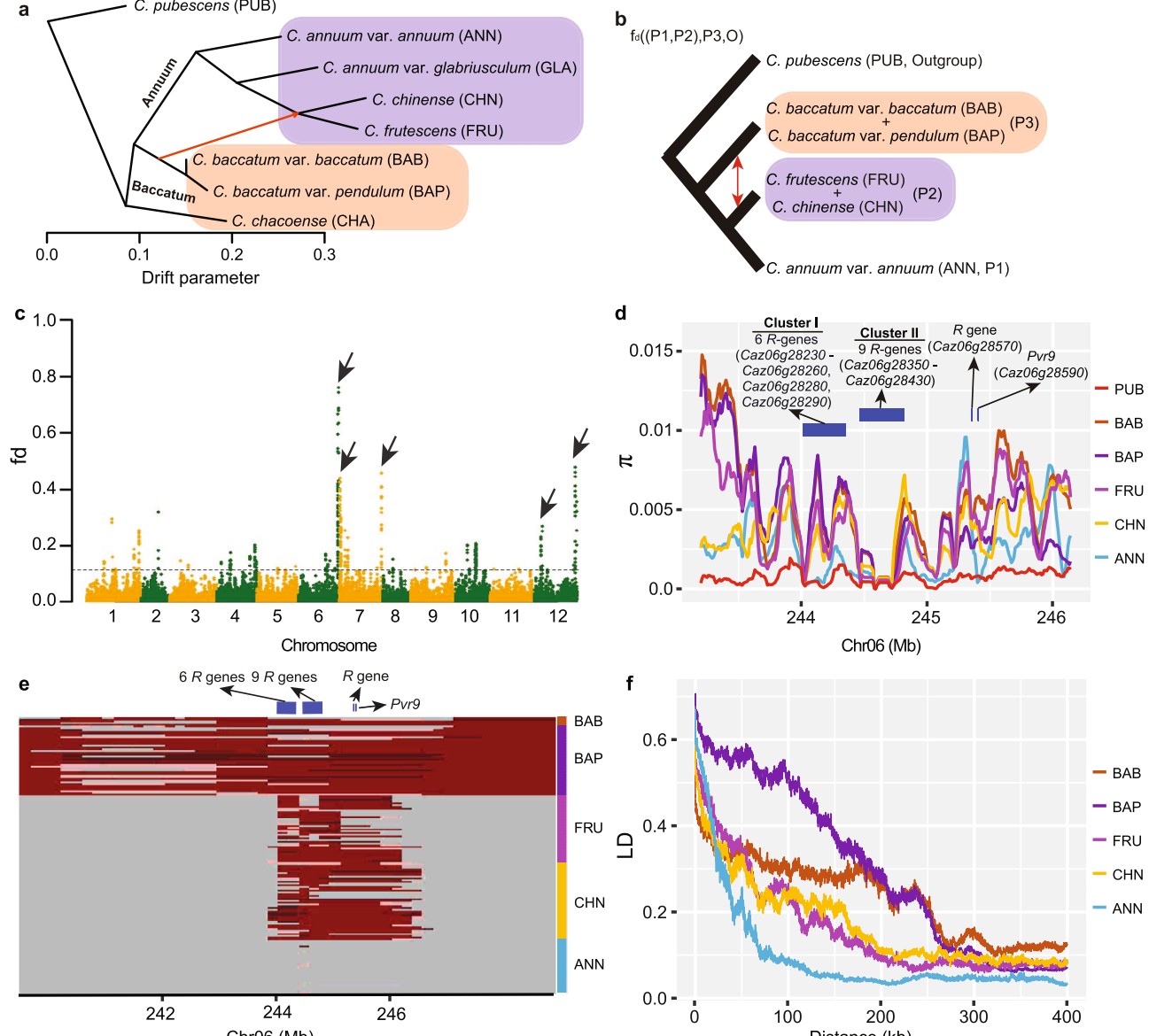

**Fig. 4 | Introgressions among the *Capsicum* species. a** Phylogenetic structure and gene flow detected between the Baccatum and the Annuum clades. The red arrow line indicates the direction of gene flow. **b** Four-taxon topology ((P1,P2),P3,O) used in the ABBA-BABA model to detect the degree of introgression (*fd*) and proportion of introgression across the whole genome (PGI) between *C. baccatum* (var. *baccatum* and var. *pendulum*) and the subgroup of the Annuum clade including *C. frutescens* and *C. chinense*. **c** Degree of introgression (*fd*) from *C. baccatum* to *C. chinense* and *C. frutescens* across the pepper genome. The top five strongest introgression signals are indicated by black arrows. Dashed horizontal lines indicate the top 1% *fd*. **d** Nucleotide diversity (π) of the genome region at the end of chromosome 6 containing the strongest introgression signal. **e** Introgression from *C. baccatum* to *C. chinense* and *C. frutescens* at the end of chromosome 6. The dark red color represents alleles prevalent in *C. baccatum* accessions (allele frequency ≥ 0.8). **f** LD decay patterns of the introgressed region shown in (**e**). PUB, *C. pubescens*; BAB, *C. baccatum* var. *baccatum*; BAP, *C. baccatum* var. *pendulum*; FRU, *C. frutescens*; CHN, *C. chinense*; ANN, *C. annuum* var. *annuum*.

independent domestications. Further large-scale phenotypic analysis of the core collection would provide additional insights into the evolution and divergence of these important domestication traits in different pepper species.

Introducing resistance genes from related species into modern commercial cultivars is an important and frequently used strategy for enhancing resistance characteristics. Our results provide evidence of introgression from *C. baccatum* into *C. chinense* and *C. frutescens*, carrying genes conferring various biotic and abiotic stress resistances (Fig. 5). The introgressed alleles from *C. baccatum* were not found in *C. annuum* var. *annuum*, which might reflect prezygotic and postzygotic barriers preventing direct crosses between *C. annuum* var. *annuum* and *C. baccatum*[44,45]. Readily crossable with *C. annuum* var. *annuum*, the *C. chinense* and *C. frutescens* accessions carrying the beneficial

*C. baccatum* alleles could be used to introduce the disease resistance traits into *C. annuum* var. *annuum*.

## Methods

### Plant materials and genome and transcriptome sequencing

A total of 1296 accessions from nine different *Capsicum* species (prior to taxonomy reclassification) were used in this study, of which 1210 were obtained from the U.S. National Plant Germplasm System and the remaining 86 were obtained from the Pepper Germplasm Bank at Hunan Academy of Agricultural Science, China. Genomic DNA was extracted from young fresh leaves of these accessions using the Plant Genomic DNA kit (Tiangen, China) following the manufacturer's instructions. A total of 1.5 μg DNA per sample was used to construct Illumina DNA libraries with insert sizes of ~350 bp, using the Truseq

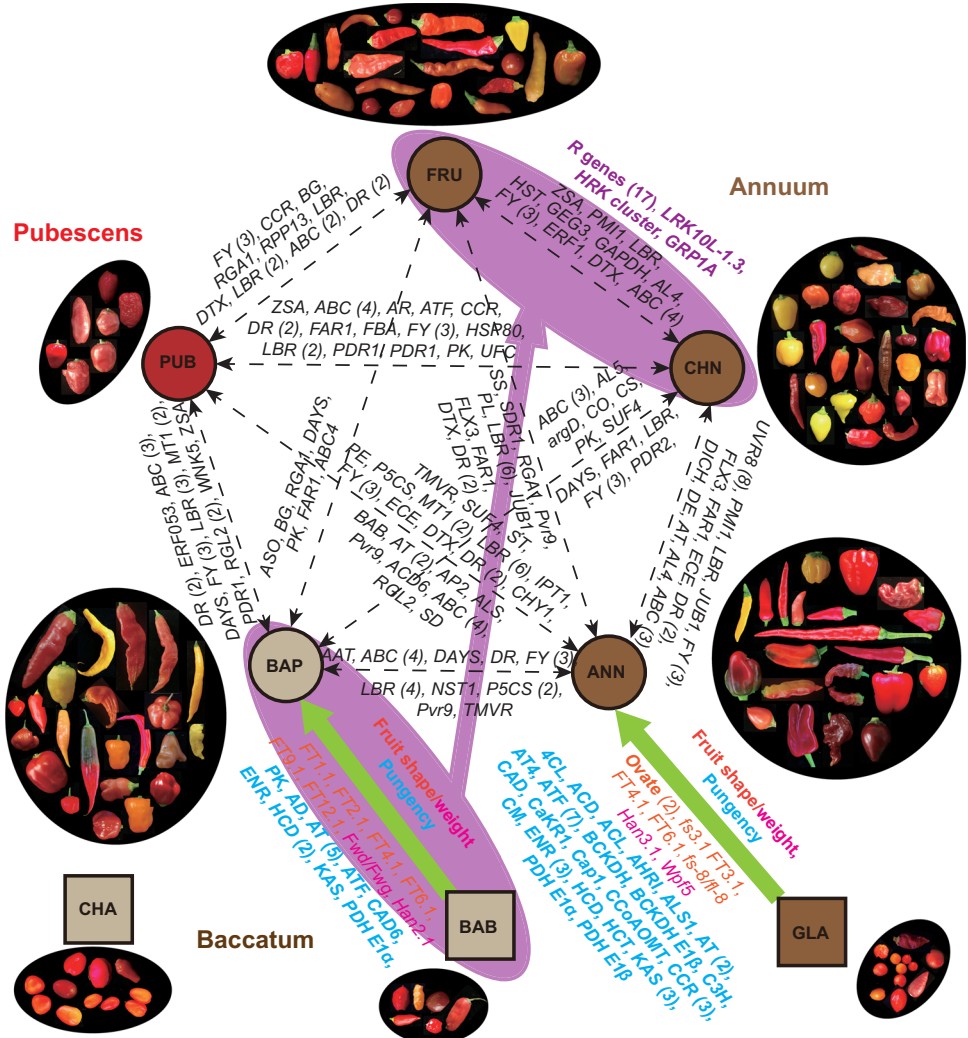

**Fig. 5 | Domestication, differentiation and introgression of the *Capsicum* species.** Representative fruits are presented for each species/group. Circles and squares represent the domesticated and wild types, respectively. Dark red, light brown and brown represent Pubescens, Baccatum and Annuum clades, respectively. Genes related to population differentiation (dashed white arrows) are indicated in white and introgression (purple arrow) are indicated in purple colors, respectively.

Selected genes (in bold) and QTLs related to fruit shape, fruit size/weight and pungency during pepper domestication (green arrows) are highlighted in pink, green and cyan, respectively. PUB, *C. pubescens*; CHA, *C. chacoense*; BAB, *C. baccatum* var. *baccatum*; BAP, *C. baccatum* var. *pendulum*; FRU, *C. frutescens*; CHN, *C. chinense*; GLA, *C. annuum* var. *glabriusculum*; ANN, *C. annuum* var. *annuum*. Genes/QTLs are the same as those shown in Figs. 2 and 3 and Supplementary Fig. 8.

Nano DNA HT Sample Preparation Kit (Illumina, USA) following the manufacturer's protocol. These libraries were sequenced on the Illumina NovaSeq 6000 platform, and paired-end reads of 150 bp in length were generated.

For genome assemblies, seedlings of *C. annuum* var. *annuum* Zhangshugang, *C. baccatum* var. *pendulum* PI 632928 and *C. pubescens* Grif 1614 were grown in a greenhouse and transferred to a dark room for 24 h before collection of young fresh leaves. High molecular weight DNA was extracted from young fresh leaves using the cetyltrimethylammonium bromide method[46] and used for the construction of the PacBio SMRT library following the standard SMRT bell construction protocol (PacBio, USA). The SMRT library was sequenced on a PacBio Sequel platform to generate CLR reads for Zhangshugang and PI 632928, and on a PacBio Sequel II platform to generate HiFi reads for Grif 1614. In addition, Illumina paired-end libraries were constructed using the Truseq Nano DNA HT Sample Preparation Kit (Illumina, USA), and Hi-C libraries were prepared following the proximo Hi-C plant protocol (Phase Genomics, USA) for the three pepper lines. Both Illumina and Hi-C libraries were

sequenced on the Illumina NovaSeq 6000 platform with the paired-end, 150-bp mode.

To assist protein-coding gene prediction, transcriptome sequencing was performed using tissues from leaf, flower and fruit. Total RNA was extracted using the TRIzol reagent (Invitrogen). Strand-specific RNA-Seq libraries were constructed using the Illumina TruSeq RNA Sample Prep Kit and sequenced on the NovaSeq 6000 platform. In addition, a pool of RNA mixed from all samples was used to construct one PacBio Iso-Seq library with the SMRTbell Express Template Prep kit 2.0 and sequenced on the PacBio Sequel II platform.

**Plant growth conditions and phenotyping**

Seeds of pepper accessions were sterilized with 5% sodium hypochlorite solution for 15 min, washed with water and then sown in a 200-hole seedling tray filled with vermiculite. The seedlings were grown in greenhouses at 25/18 °C day/night temperature and with a 16/8-h light/dark cycle, relative humidity of 60–70%, and light intensity of 15000 Lux at Changsha (N28°15′10″, E113°5′9″), Hunan province, China. Three ripe fruits were harvested per plant, and each fruit was photographed,

and the lateral and longitudinal lengths of the fruit were measured with three replicates.

## De novo genome assembly

For Zhangshugang and PI 632928, PacBio CLR reads were error-corrected and de novo assembled into contigs using MECAT2 (ref. [47]) with default parameters. By aligning the PacBio reads to the assembled contigs, errors in the assembled contigs were corrected using the Arrow pipeline from the SMRT link 4 toolkit (https://www.pacb.com/products-and-services/analytical-software/smrt-analysis/). The assembled contigs were further polished with the Illumina short reads using Pilon[48] (v1.22). For Grif 1614, PacBio HiFi reads were assembled into contigs using Hifiasm[49] (v0.16.0) with default parameters. Purge Haplotigs[50] (v1.1.2) was then used to remove redundancies in the assembled contigs with default parameters. Assembled contigs were also compared against the NCBI non-redundant nucleotide database to remove possible contaminated sequences from organelle and microorganism genomes. To build pseudomolecules from the assembled contigs, Hi-C reads were cleaned to remove adapters and low-quality sequences using Trimmomatic[51] (v0.39), and then aligned to the contigs using BWA[52] (v0.7.17) with default parameters. The contigs were then clustered into pseudomolecules using the agglomerative hierarchical clustering method implemented in LACHESIS[53]. Base accuracy and completeness of the genome assemblies were evaluated using Merqury[12] with default settings and BUSCO[13] using the embryophyta odb10 database, respectively.

## Repetitive sequence annotation and protein-coding gene prediction

Repetitive sequences in the three genomes were identified by integrating homology-based and de novo predictions. For homology-based prediction, transposable elements (TEs) in Repbase[54] were used to scan the genomes with RepeatMasker[55] (v4.0.9). The de novo prediction was performed using RepeatModeler (http://www.repeatmasker.org/RepeatModeler/), which is based on de novo repeat detection programs, RECON[56] (v1.08) and RepeatScout[57] (v1.0.5). Furthermore, a de novo search for long terminal repeat (LTR) retrotransposons in the genomes was performed using LTR_FINDER[58] (v1.0.7).

Protein-coding genes were predicted from the repeat-masked genomes by integrating evidence from three different approaches: ab initio, homology-based and transcriptome-based gene predictions. First, RNA-Seq reads from different tissues and developing stages were aligned to the genome assemblies using HISAT2 (ref. [59]) (v2.0.4) and then assembled into transcripts using StringTie[60] (v2.0). These transcripts, combined with the open reading frames predicted from the Iso-Seq transcripts using PASA[61] (v2.0.1), were used as transcript evidence for gene prediction. AUGUSTUS[62] (v3.3.1) and GENESCAN[63] were used to perform ab initio gene prediction. A set of high-quality full-length cDNA sequences derived from the transcriptome sequences were used to train the ab initio gene predictors. Exonerate[64] (v2.2.0) was used for homology-based gene prediction with protein sequences from the four Capsicum genomes, C. annuum var. annuum CM334, C. baccatum PBC81, C. chinense PI 159236 and C. annuum var. annuum Zunla-1 (refs. [7,8]). Finally, prediction results from the three approaches were integrated by MAKER[65] (v3.00) to derive a set of consensus gene models for each of the three genome assemblies. The predicted protein-coding genes were functionally annotated by comparing their protein sequences against the NCBI non-redundant (nr), TrEMBL/SwissProt[66] and InterPro domain[67] databases. Gene ontology (GO) terms were assigned to predicted genes using Blast2GO[68].

## Graph pan-genome construction

A graph pan-genome was constructed using the Zhangshugang genome as the backbone. Genome sequences of PI 632928 and Grif 1614 were aligned to the Zhangshugang genome using minimap2[69] (v2.23)

with the parameter '-asm20'. Small indels and large SVs were then identified using the paftools in the minimap2 package with the parameter '-L 10000'. In addition, genome sequences of PI 632928 and Grif 1614 were also aligned to the Zhangshugang genome using MUMmer[70] (v4.0.0) with the parameter '--maxmatch', and based on the alignments SVs were identified using Assemblytics[71] (v1.2.1) with unique anchor length set to 5000 bp, minimum and maximum SV lengths set to 20 bp and 2 Mb, respectively. SVs identified from paftools and Assemblytics were then merged using bcftools[72] (v1.14), and inconsistent SVs called from the two programs were removed. The Zhangshugang reference genome and the identified small indels and SVs were used to build the graph pan-genome using the vg toolkit[73] (v1.40.0).

## SNP and small indel calling for the pepper collection

Raw Illumina DNA reads were processed to remove adapter and low-quality sequences using Trimmomatic[51] (v0.39) with parameters "TruSeq3-PE-2.fa:2:30:10:1:TRUE SLIDINGWINDOW:4:20 LEADING:3 TRAILING:3 MINLEN:40". The cleaned reads were then mapped to the graph pan-genome using the Giraffe mapper[74] with default parameters. The resulting aligned gam files were converted to bam files using 'vg surject'. SNPs and small indels were then called from bam files using the Sentieon software package (https://www.sentieon.com/), which was built based on the GATK variant calling tool[75]. Briefly, duplicated read pairs in each alignment file were marked, and indel realignment and base quality score recalibration were performed. Variants from each sample were called using the Sentieon Haplotyper function, and joint variant calling was performed using the GVCFtyper function in Sentieon. Raw SNPs were then filtered using GATK[75] (v3.8) with parameters 'QD < 2.0||FS > 60.0||MQ < 40.0|| SOR > 3.0 || MQRankSum < −12.5 || ReadPosRankSum < −8.0' and raw small indels were filtered with parameters 'QD < 2.0||FS > 200.0 || ReadPosRankSum < −20.0'. Filtered bi-allelic variants with minor allele frequency (MAF) ≥ 0.05 and missing data rate ≤ 0.3 were kept. Variants were annotated using SnpEff[76] (v5.0e). SNPs were further LD-pruned with parameters '--indep-pairwise 50 10 0.3' using PLINK (v1.9)[77], and the LD-pruned SNPs were used for downstream population genomic analyses unless otherwise indicated.

## Phylogenetic and population genomic analyses

A total of 28,980 LD-pruned SNPs at fourfold degenerate (4DTv) sites were used to construct the maximum likelihood (ML) phylogenetic tree and to perform population structure and principal component analysis (PCA). The ML phylogenetic tree was constructed using IQ-TREE[78] (v1.6.12) with 1000 bootstrap replicates and the optimal substitution model of TVM+F+R4, which was determined based on the lowest BIC (Bayesian Information Criterion) value using ModelFinder[79]. Four wild tomato accessions, Solanum pennellii LA1272, S. pennellii LA0716, S. habrochaites LA1777 and S. habrochaites LA1718, were used as the outgroup. Illumina reads of these four accessions were downloaded from NCBI Sequencing Read Archive under accession numbers ERR418106, ERR418107, ERR418103 and ERR418102, and aligned to the pepper graph pan-genome for SNP calling as described above. The population structure of the Capsicum accessions was inferred using fastSTRUCTURE[80] (v1.0) with the number of population clusters (K) set from 2 to 20. The optimal K (7) was determined using the chooseK.py script in fastSTRUCTURE. PCA was performed using PLINK[77] (v1.9). Accessions with significant genome-wide hybridization were identified using HyDe[81] with a P-value cutoff of 0.05 and were excluded from the downstream population genomic analyses (Supplementary Data 4).

Genome-wide LD decay pattern for each of the Capsicum groups was calculated using PopLDdecay[82] (v3.41) with the parameter '-MaxDist 1000'. $F_{ST}$ values between different Capsicum groups were calculated using VCFtools (v0.1.17) with a window size of 100 kb and a step size of 10 kb. To evaluate the significance of population differentiation, for each pair of compared populations, 100 $F_{ST}$ values were generated from

permutations through randomly reshuffling the accessions and these values were compared to the empirical $F_{ST}$ value. Highly differentiated genomic regions between different *Capsicum* groups were identified as windows with top 1% $F_{ST}$ values, and windows were merged if they overlapped. Genome-wide nucleotide diversity (π) was calculated in 100-kb non-overlapping windows using VCFtools[83]. The significance of the mean π values between *Capsicum* groups was calculated using the two-tailed t-test with the Bonferroni adjust method using the pairwise.t.test() function in the R package stats (v3.6.2).

### Inference of effective population size

The effective population size (*Ne*) for each of the eight *Capsicum* groups was inferred using SMC++[84] (v1.15.4.dev18+gca077da) with genome-wide SNPs. Ten accessions from each group from different representative origins were selected for the analysis. SNP missing rate was set to less than 0.2, and the runs of homozygosity longer than 5 kb were treated as missing to avoid potential bias. In order to improve the estimation accuracy, we used the composite likelihood approach by setting each of the 10 accessions as the 'distinguished individual' and the remaining nine as 'undistinguished individuals'. The mutation rate and generation time were set at $6.96 \times 10^{-9}$ (ref. 85) and one year, respectively.

### Identification of selective sweeps

Selective sweeps across the pepper genome in the cultivated *C. annuum* var. *annuum* population and in the cultivated *C. baccatum* var. *pendulum* population were identified using SweeD[86] (v4.0.0) with genome-wide SNPs, by incorporating the empirical estimate of the effective population size derived from the SMC++ analyses described above. Composite likelihood ratios (CLR) were calculated in an average of a 10-kb window across the genome by setting grid numbers according to chromosome lengths (number of grid = chromosome length/10000). Those with the top 5% highest CLR values were identified as potential selective sweeps and sweeps with physical distance no larger than 10 kb were merged. Furthermore, π ratios between the wild *C. annuum* var. *glabriusculum* and the cultivated *C. annuum* var. *annuum*, and between the wild *C. baccatum* var. *baccatum* and the cultivated *C. baccatum* var. *pendulum* were calculated. Potential selective sweeps identified from SweeD having the top 30% of π ratios between the wild and cultivated species were kept as the final selective sweeps.

### Genome-wide association study

Genome-wide SNPs were used for GWAS analysis using EMMAX[87]. Phenotype data of fruit shape-related traits, including fruit length, fruit width and fruit width/length ratio, were log2 transformed. The top 5 principal components and the Balding-Nichols kinship were used as cofactors in the association model. The effective number of SNPs was calculated using Genetic Type I error Calculator[88]. The Bonferroni-corrected *P*-value thresholds were $2.34 \times 10^{-8}$ at Type I error α = 0.05 (highly significant) and $4.68 \times 10^{-7}$ at α = 1 (significant), respectively, for *C. annuum*, corresponding to $-\log_{10}(P)$ values of 7.63 and 6.33, and $2.13 \times 10^{-8}$ (α = 0.05) and $4.25 \times 10^{-7}$ (α = 1) for *C. baccatum*, corresponding to $-\log_{10}(P)$ values of 7.67 and 6.37. The candidate gene region was determined by LD between the peak SNP and its nearby SNPs with the LD cutoff of 0.351 and 0.366 (half value of the maximum averaged LD) for *C. annuum* var. *annuum* and *C. baccatum* var. *pendulum*, respectively.

### Detection of gene flow and introgressions among the *Capsicum* species

Gene flow between the eight *Capsicum* groups was detected using TreeMix[39] (v1.13). To obtain the optimal number of migration edges, the number of migrations was set from 0 to 20, each with 10 interactions. The optimal number of migration edges (m = 1) was obtained

using the linear modeling estimate in the OptM R package (v0.1.6; https://cran.r-project.org/web/packages/OptM), where negligible variance was further explained with the increase of migration (Supplementary Fig. 7).

Genome-wide introgressions were detected by calculating the $f_d$ statistics[40] in 500-kb sliding windows with a step size of 50 kb, using genome-wide SNPs. The minimum number of SNPs per window was set to 100, and the minimum proportion of samples genotyped per site was set to 0.5. We further calculated the PGI (proportion of introgression across the whole genome), following the recently proposed formula[89]:

$$\text{PGI} = \frac{\sum f_{di} \times G_i}{G} \tag{1}$$

where PGI refers to the average proportion of introgression across the whole genome; $f_{di}$ refers to the $f_d$ value of the $i_{th}$ window; $G_i$ refers to the $i_{th}$ window size in base pairs; $G$ refers to the total genome size in base pairs.

### Reporting summary

Further information on research design is available in the Nature Portfolio Reporting Summary linked to this article.

## Data availability

Genome assemblies, raw genome and transcriptome sequencing reads have been deposited in the National Center for Biotechnology Information BioProject database under the accession nos. PRJNA800056 and PRJNA801499. Genome assemblies and annotated genes, pangenome graphs (Giraffe indexes), and variant data (VCF format) are also available at the Pepper Genomics Database (http://ted.bti.cornell.edu/pepper). Source data are provided with this paper.

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

## Acknowledgements

This research was supported by grants from the Laboratory of Lingnan Modern Agriculture Project (NT2021004), the National Natural Science Foundation of China (U19A2028, U21A20230 and 32130097), the Science and Technology Innovation Program of Hunan Province, China (2021NK1006 and 2021JC0007), and the US National Science Foundation (IOS-1855585 to Z.F.).

## Author contributions

X.Z., Z.F. and S.W. designed and managed the project. F.L., C.X., Z.W., J.W., B.T., H.X., B.H., H.Suo, B.Y., L.O., X.L., S.Z., S.Y., Z.L., F.Y., Z.P., Y.M. and X.D. collected samples and performed experiments. J.Z., F.L., H.Sun, X.S., X.W. and R.J. performed data analyses. J.Z. and S.W. wrote the manuscript. Z.F. revised the manuscript.

## Competing interests

The authors declare no competing interests.
