## [Peer Review File · Nature Communications]

Genomes of cultivated and wild Capsicum species provide insights into pepper domestication and population differentiationEditorial Note: This manuscript has been previously reviewed at another journal that is not operating a transparent peer review scheme. This document only contains reviewer comments and rebuttal letters for versions considered at Nature Communications.

Reviewers' Comments:

Reviewer #5:

Remarks to the Author:

I think that the manuscript in the present review phase responds to the requests and observations made to the authors by the reviewers and that the weaknesses have been resolved by the authors themselves.

Furthermore, the work provides useful information in the study of *Capsicum* spp and Solanaceae in general, as well as proposing a pangenome graph of three different species.

There are just a few points that need attention:

- 1) line 93: I find figures S1 and S2 not in line with the text to which they refer. S1 reports the Mercury analysis, S2 represents the HI-Cc interaction map.
- 2) line 95 Busco and Mercury are missing in Material and methods. I suggest to add them
- 3) Line 208. Why did the authors selected a LD value of 0.3 for defining the threshold?
- 4) line 311 and 313, reference are bad written (ref32)
- 5) line 332. Author says the "However, GWAS of fruit shape index in *C. annum* identified a strong signal on chromosome 2 between 160,915,869 bp and 333 161,006,859 bp". Looking at the figure 2f, the signal seems not very strong. How did the authors define "strong" a signal? Based on pvalue? at which threshold. The same for Figure S6, line 331
- 6) line 359: In this section, just one migration event is reported for Treemix, while in material and methods, line 627, 5 migrations are suggested.

Reviewer #6:

Remarks to the Author:

Dear Dr. An,

I have read the Liu et al. manuscript, the reviews of the previous submission, and the authors' response to the reviews. There are a couple of minor issues, but on the whole I think the authors have addressed the issues raised in the reviews. The couple of issues.

1. Reviewer #2 asked the authors to use permutations to generate null distributions that could be used to assign P-values to the F_{st} values of the pairs of subspecies. This was a reasonable ask by the reviewer. The authors do now report a P-value (2.2×10^{-16}). However, this value is not meaningful for a few reasons. First, the authors do not explain what this is a probability of (my guess is some sort of t-test of whether the means differ). Second, there are only 100 permutations (and one observed value), so it is impossible to have a P-value $< 2.2 \times 10^{-16}$. Using a t-test (or really any statistical test) makes no sense with permuted data (the value of P can be manipulated by simply increasing the number of permutations). Moreover, only a single genome wide F_{st} value is reported (I assume this is mean, but they don't say).

What the authors should do is report the number (or proportion) of permutations which have a F_{st} value that is higher than the empirical F_{st} value(s). Finally, it is not clear how the permutations were done – they should be pairwise, as written in the Methods it suggests that accessions were included for each permutation.

Of course, these permutations do not provide any insight into the probability that the 1% of top windows harbor F_{st} values greater than expected by chance. That is a different issue, some people are bothered by this and would like to see coalescent simulations to provide P values. I personally am not particularly concerned and think examining the top 1% is a reasonable thing to do.

2. The authors selected a set of 500 accessions for analyses from an initial collection of 1296 accessions. This subset of accessions was selected by removing accessions that had highly similar genomes (on the bases of shallow genome sequencing). This makes total sense for the GWAS analyses and makes total sense for maximizing the number of genomic variants identified. However, sub-setting samples to increase diversity is a problem for demographic inferences as well as estimates of diversity. In other words, there is an ascertainment bias that increases diversity, and biases estimates of π and thus population size. As the data are collected, I am not sure what to do about this, but it seems like some genomic simulations might allow one to estimate the magnitude of the bias. Certainly one shouldn't simply ignore the bias when they report the estimates of diversity and demographic history.

3. Related to the π values, the methods state the a t-test was used to compare π values among groups. I don't know the specific R package they used, but a couple of issues. First, what was the N used? A step size one tenth of the window size was used when calculating π , if all values were then used for the t-test, then there is a problem because each section of the genome has been counted 10, instead of a single time. Second, Figure 1g reports to indicated π estimates that differ using letters....but t-tests are pairwise tests? So what are the results, that all species estimates differed from all other species estimates? If so, seems silly to have a different letter on each species in the figure. Moreover, the authors us a P value < 0.05, but with pairwise tests among 8 taxa there is a substantial multiple testing problem.

4. Minor, but there is no reason for figure 2b (total length of the genomic windows that harbor the top 1% of F_{st} values) – because windows were the same size for each species pair this must be just 1% of the genome.

Reviewer #5 (Remarks to the Author):

I think that the manuscript in the present review phase responds to the requests and observations made to the authors by the reviewers and that the weaknesses have been resolved by the authors themselves.

Furthermore, the work provides useful information in the study of *Capsicum spp.* and Solanaceae in general, as well as proposing a pangenome graph of three different species.

There are just a few points that need attention:

1) line 93: I find figures S1 and S2 not in line with the text to which they refer. S1 reports the Merqury analysis, S2 represents the HI-C interaction map.

Response: Thanks for pointing this out. The two figures have now been cited in right places.

2) line 95 Busco and Merqury are missing in Material and methods. I suggest to add them

Response: Done (Line 503-505). Thanks.

3) Line 208. Why did the authors selected a LD value of 0.3 for defining the threshold?

Response: An r^2 of 0.3 corresponded to the half of their maximum LD values for several pepper populations. Nonetheless, this is somewhat arbitrary (e.g., 0.2 was used in <https://www.nature.com/articles/ng.2313>). **Fig. 2f** shows the LD decay patterns of different cultivated and wild pepper populations, in which *C. annuum* var. *annuum* displayed the slowest LD decay. We now think it is not necessary to indicate the physical distances at LD of 0.3; therefore, we have removed the dotted lines and physical distances at $r^2=0.3$ in **Fig. 2f**.

4) line 311 and 313, reference are bad written (ref32)

Response: These are correct citation format according to the style of Nature Communications.

5) line 332. Author says the “However, GWAS of fruit shape index in *C. annuum* identified a strong signal on chromosome 2 between 160,915,869 bp and 333 161,006,859 bp”. Looking at the figure 2f, the signal seems not very strong. How did the authors define "strong" a signal? Based on pvalue? at which threshold. The same for Figure S6, line 331

Response: We have changed “a strong signal” to “a highly significant association signal”. In this study, the Bonferroni-corrected thresholds for the P values were 2.34×10^{-8} at $\alpha = 0.05$ (highly significant) and 4.68×10^{-7} at $\alpha = 1$ (significant), respectively, for *C. annuum*, with corresponding $-\log_{10}(P)$ values of 7.63 and 6.33. We have added this to the Methods (Line 623-626).

6) line 359: In this section, just one migration event is reported for Treemix, while in material and methods, line 627, 5 migrations are suggested.

Response: We thank the reviewer for pointing this out. It was a typo on line 627 (Line 634 in the revised manuscript). The optimal number of migrations estimated by the optM R package was one as shown in Supplementary Figure 7. We have corrected this error in the text.

Reviewer #6 (Remarks to the Author):

Dear Dr. An,

I have read the Liu et al. manuscript, the reviews of the previous submission, and the authors' response to the reviews. There are a couple of minor issues, but on the whole I think the authors have addressed the issues raised in the reviews. The couple of issues.

1. Reviewer #2 asked the authors to use permutations to generate null distributions that could be used to assign P -values to the F_{ST} values of the pairs of subspecies. This was a reasonable ask by the reviewer. The authors do now report a P -value (2.2×10^{-16}). However, this value is not meaningful for a few reasons. First, the authors do not explain what this is a probability of (my guess is some sort of t-test of whether the means differ). Second, there are only 100 permutations (and one observed value), so it is impossible to have a P -value $< 2.2 \times 10^{-16}$. Using a t-test (or really any statistical test) makes no sense with permuted data (the value of P can be manipulated by simply increasing the number of permutations). Moreover, only a single genome wide F_{ST} value is reported (I assume this is mean, but they don't say).

What the authors should do is report the number (or proportion) of permutations which have a F_{ST} value that is higher than the empirical F_{ST} value(s). Finally, it is not clear how the permutations were done – they should be pairwise, as written in the Methods it suggests that accessions were included for each permutation.

Of course, these permutations do not provide any insight into the probability that the 1% of top windows harbor F_{ST} values greater than expected by chance. That is a different issue, some people are bothered by this and would like to see coalescent simulations to provide P values. I personally am not particularly concerned and think examining the top 1% is a reasonable thing to do.

Response: We appreciate the reviewer's comment that examining the top 1% windows is a reasonable thing to do. For the permutation tests for significant population differentiation of each comparison, we randomly shuffled the accessions to obtain 100 F_{ST} values and compared them to the empirical F_{ST} value. We have added this to the Methods (Line 583-587). We agree with the reviewer that reporting the number or proportion of F_{ST} values from permutations higher than the empirical value is a better indication of whether the two populations are significantly differentiated. For each of the comparisons shown in Fig. 2, all 100 F_{ST} values generated from permutations were lower than the empirical F_{ST} value (**Response Table 1**); this is also why we have obtained very low p values. We have added this information in the legend of Fig. 2.

Response Table 1. F_{ST} values from 100 permutations

GroupA	GroupB	Minimum	Maximum	Mean	Empirical F_{ST}
PUB	CHA	0.014510	0.090313	0.033876	0.626321
PUB	BAB	0.027504	0.170543	0.054837	0.565119
PUB	BAP	0.006982	0.049013	0.012079	0.578143
PUB	GLA	0.014048	0.123794	0.030655	0.533744
PUB	ANN	0.005464	0.073070	0.013026	0.535831
PUB	FRU	0.005897	0.055068	0.012506	0.658525
PUB	CHN	0.006216	0.042777	0.010246	0.566640
CHA	BAB	0.034268	0.167742	0.056443	0.490290
CHA	BAP	0.013249	0.084802	0.028916	0.533545
CHA	GLA	0.019491	0.158323	0.035331	0.519987
CHA	ANN	0.012247	0.147721	0.022255	0.521746
CHA	FRU	0.013025	0.048124	0.020425	0.626814

CHA	CHN	0.009546	0.039228	0.018336	0.588533
BAB	BAP	0.029599	0.056811	0.039804	0.292106
BAB	GLA	0.024995	0.089884	0.038348	0.471852
BAB	ANN	0.018743	0.115261	0.031184	0.477183
BAB	FRU	0.019589	0.150278	0.034576	0.564351
BAB	CHN	0.021344	0.055268	0.029984	0.551821
BAP	GLA	0.011726	0.144979	0.026563	0.466169
BAP	ANN	0.003208	0.059443	0.007272	0.452700
BAP	FRU	0.002582	0.020801	0.005349	0.604932
BAP	CHN	0.002668	0.018769	0.004984	0.499586
GLA	ANN	0.009354	0.129885	0.020249	0.562410
FRU	CHN	0.003396	0.016458	0.005656	0.371065
FRU	GLA	0.008496	0.062431	0.016744	0.498129
FRU	ANN	0.003008	0.035223	0.006658	0.562410
CHN	GLA	0.009373	0.107650	0.016794	0.505557
CHN	ANN	0.002942	0.021287	0.005699	0.478674

2. The authors selected a set of 500 accessions for analyses from an initial collection of 1296 accessions. This subset of accessions was selected by removing accessions that had highly similar genomes (on the bases of shallow genome sequencing). This makes total sense for the GWAS analyses and makes total sense for maximizing the number of genomic variants identified. However, sub-setting samples to increase diversity is a problem for demographic inferences as well as estimates of diversity. In other words, there is an ascertainment bias that increases diversity, and biases estimates of π and thus population size. As the data are collected, I am not sure what to do about this, but it seems like some genomic simulations might allow one to estimate the magnitude of the bias. Certainly one shouldn't simply ignore the bias when they report the estimates of diversity and demographic history.

Response: We understand the reviewer's concern that nucleotide diversity could be affected by removing individuals with more similar genetic backgrounds when subsetting accessions from 1,296 to 500. We have attempted to use the SNPs derived from shallow whole genome sequencing data ($\sim 1\times$) to investigate the effect of sub-sampling on nucleotide diversity estimation, but found out that unfortunately the $1\times$ sequencing data were not suitable for this analysis due to the very low coverage. Nonetheless, when we constructed the 500-accession core using GenoCore, the first 50 and 100 accessions already captured 91.3% and 92.9% of the genetic diversity in the entire population, respectively, while the remaining 400 accessions only added 4.5% of the genetic diversity. This indicates that the selection of the majority of the accessions in the 500-core was close to random. Therefore, we think although the sub-sampling could overestimate the nucleotide diversity, the effect should be minor and will not affect the overall conclusion of our study. We have added a note in the legend of Fig. 2 to acknowledge that the π values calculated from the core collection could be an overestimation for the 1,296 accessions (Line 208-209).

We don't think sub-sampling would affect the inference of effective population size as the method used for this analysis is not related to sub-sampling. As indicated in the Methods, for each group we selected ten accessions from different representative origins to infer the effective population size using the SMC++ package (Line 595-602).

3. Related to the π values, the methods state that a t-test was used to compare π values among groups. I don't know the specific R package they used, but a couple of issues. First, what was the N used? A step size one tenth of the window size was used when calculating π , if all values were then used for the t-test, then there is a problem because each section of the genome has been counted 10, instead of a single time. Second, Figure 1g reports to indicated π estimates that differ using letters....but t-tests are pairwise tests? So what are the results, that all species estimates differed from all other species estimates? If so, seems silly to have a different letter on each species in the figure. Moreover, the authors us a P value < 0.05 , but with pairwise tests among 8 taxa there is a substantial multiple testing problem.

Response: Thanks for pointing this out. In the revised manuscript, we calculated π values using **non-overlapping** 100-kb windows ($N = 30,187$), instead of using a window-step approach. A two-tailed t-test was performed for each pair-wise comparison followed by Bonferroni correction for multiple testing. All the pair-wise comparisons showed significantly different means (adjusted P -value < 0.05) of the two groups. We have made this clear in the revised manuscript (Line 589-592). Nonetheless, we have removed the letters from Fig. 2g, as the boxplots alone are sufficient to provide the necessary information.

4. Minor, but there is no reason for figure 2b (total length of the genomic windows that harbor the top 1% of F_{ST} values) – because windows were the same size for each species pair this must be just 1% of the genome.

Response: We are sorry for the confusion. F_{ST} values were calculated for 100-kb windows with a step size of 10 kb. This window-step approach, instead of using non-overlapping windows, allowed us to identify more accurate locations (boundaries) of the highly differentiated genomic regions. Windows with top 1% values were considered significantly differentiated and merged if they overlapped. Therefore, the total size of significantly differentiated genomic regions could be different among different species pairs, and was smaller than 1% of the genome due to merging of overlapping windows. We have made this clear in the revised manuscript (Line 581-583 and 587-589)

Reviewers' Comments:

Reviewer #5:

Remarks to the Author:

All my concerns have been addressed by authors.

Reviewer #6:

Remarks to the Author:

The authors have adequately addressed the issues I raised in my review of the previous version of this manuscript. That said,

1. I don't understand why the authors continue to use a t-test to compare their empirical estimate to the values from the permutations – it would be better to just report the number of permutations that were less (or greater) than the empirical value.

2. The authors are likely correct that subsampling the data prior to estimating π will have had little effect. But, they really haven't shown that. Moreover, their response is focused on finding common variants – it is not common variants that are an issue (recovering common variants will not be affected by pruning a sample on the basis of close similarity), it is the rare variants that are the issue.

Reviewer #5:

All my concerns have been addressed by authors.

Response: Thanks.

Reviewer #6:

The authors have adequately addressed the issues I raised in my review of the previous version of this manuscript. That said,

1. I don't understand why the authors continue to use a t-test to compare their empirical estimate to the values from the permutations – it would be better to just report the number of permutations that were less (or greater) than the empirical value.

Response: A t-test was suggested by a previous reviewer. We agree with the reviewer and have now deleted the sentences related to the t-test.

2. The authors are likely correct that subsampling the data prior to estimating π will have had little effect. But, they really haven't shown that. Moreover, their response is focused on finding common variants – it is not common variants that are an issue (recovering common variants will not be affected by pruning a sample on the basis of close similarity), it is the rare variants that are the issue.

Response: We thank the reviewer for the comment. As indicated in our previous response and in the manuscript, the core collection captures ~97.5% of the total genetic diversity in the initial 1,296 *Capsicum* accessions. Therefore, the percentage of rare variants not captured in the core collection should be very low. Thus, while we agree with the reviewer that incomplete capture of rare variants by the core collection could affect the estimation of the nucleotide diversity, the effect should be minor and will not affect the overall conclusion of our study.